# Genome-wide binding analysis of 195 DNA binding proteins reveals "reservoir" promoters and human specific SVA-repeat family regulation

**Michael J. Smallegan**[1,2]*, **Soraya Shehata**[1,3,4º], **Savannah F. Spradlin**[3,4º], **Alison Swearingen**[2,4º], **Graycen Wheeler**[3,4º], **Arpan Das**[2,4], **Giulia Corbet**[1,3,4], **Benjamin Nebenfuehr**[2,4], **Daniel Ahrens**[2,4], **Devin Tauber**[1,3,4], **Shelby Lennon**[3,4], **Kevin Choi**[2,4], **Thao Huynh**[1,3,4], **Tom Wieser**[3,4], **Kristen Schneider**[1,4,5], **Michael Bradshaw**[1,4,5], **Joel Basken**[6], **Maria Lai**[6], **Timothy Read**[6], **Matt Hynes-Grace**[1], **Dan Timmons**[1], **Jon Demasi**[1], **John L. Rinn**[1,2,3]*

1 BioFrontiers Institute, University of Colorado Boulder, Boulder, Colorado, United States of America,
2 Department of Molecular and Cellular Biology, University of Colorado Boulder, Boulder, Colorado, United States of America, 3 Department of Biochemistry, University of Colorado Boulder, Boulder, Colorado, United States of America, 4 Biochemistry 5631 Spring 2020, University of Colorado Boulder, Boulder, Colorado, United States of America, 5 Department of Computer Science, University of Colorado Boulder, Boulder, Colorado, United States of America, 6 Arpeggio Biosciences, Boulder, Colorado, United States of America

☯ These authors contributed equally to this work.
* michael.smallegan@colorado.edu (MS); john.rinn@colorado.edu (JR)

**Data Availability Statement:** Accession numbers for data retrieved from ENCODE are available in S1 Table as well as in the code used to retrieve the

## Abstract

A key aspect in defining cell state is the complex choreography of DNA binding events in a given cell type, which in turn establishes a cell-specific gene-expression program. Here we wanted to take a deep analysis of DNA binding events and transcriptional output of a single cell state (K562 cells). To this end we re-analyzed 195 DNA binding proteins contained in ENCODE data. We used standardized analysis pipelines, containerization, and literate programming with R Markdown for reproducibility and rigor. Our approach validated many findings from previous independent studies, underscoring the importance of ENCODE's goals in providing these reproducible data resources. We also had several new findings including: (i) 1,362 promoters, which we refer to as 'reservoirs,' that are defined by having up to 111 different DNA binding-proteins localized on one promoter, yet do not have any expression of steady-state RNA (ii) Reservoirs do not overlap super-enhancer annotations and distinct have distinct properties from super-enhancers. (iii) The human specific SVA repeat element may have been co-opted for enhancer regulation and is highly transcribed in PRO-seq and RNA-seq. Collectively, this study performed by the students of a CU Boulder computational biology class (BCHM 5631 – Spring 2020) demonstrates the value of reproducible findings and how resources like ENCODE that prioritize data standards can foster new findings with existing data in a didactic environment.

data S1 File. The PRO-seq data has been deposited in GEO with the accession GSE166658.

**Funding:** Arpeggio Biosciences provided support in the form of salaries for authors JB, ML, and TR but did not have any additional role in the study design, data collection and analysis, decision to publish, or preparation of the manuscript. The specific roles of these authors are articulated in the 'author contributions' section. This course is supported by the CU Boulder Biochemistry department through these core training grants specifically for authors TW, SFS, TH, and GW by the Biophysics training program (T32GM065103), Signaling and cellular regulation training program (T32GM008759).

**Competing interests:** JB, ML, and TR are employed by Arpeggio Biosciences. This does not alter our adherence to PLOS ONE policies on sharing data and materials. There are no patents or products associated with this research. None of the other authors have any competing interests to declare.

## Introduction

In the postgenomic era [1,2] there have been efforts to adapt classical biochemical protocols studying a few DNA regions to genome-wide events. One of the first of these genome-wide assays was Chromatin Immunoprecipitation (ChIP) followed by hybridization of co-precipitate DNA fragments to microarrays (or ChIP-CHIP) representing many thousands of DNA locations (e.g., promoters). This application was first demonstrated in yeast and quickly adapted to many species [3–7]. With the advent of massively parallel sequencing technologies, bound DNA from the biochemical ChIP could be sequenced (ChIP-seq) to unbiasedly detect binding events (reviewed [8,9]). This rapid change in platforms for ChIP analyses resulted in many data sets that differed greatly in their results (ChIP-CHIP versus ChIP-seq) [10,11]). Only three years after sequencing of the human genome it became clear that uniform experimental and data standards were essential to limit a deluge of irreproducible results.

To this end, the field turned to the publicly available ENCODE consortium as the largest and most standardized repository of ChIP-seq data sets [12–15]. The goal was to develop standardized experimental and computational pipelines. Over the past 17 years since its inception, many thousands of ChIP-seq experiments have been performed. Often these large consortium studies analyze these data sets across cell types and tissues [13,16–19]. In contrast, fewer studies have investigated dozens of DNA binding proteins (DBPs) in one cell type.

Observing how hundreds of DBPs are bound relative to each other in the same cellular context provides a unique perspective. This allows a promoter-centric approach across hundreds of possible DNA binding events. Thus, we can address the underlying properties of combinatorial binding at promoters and, in turn, how this relates to promoter activity. Moreover, this approach allows us to systematically investigate numerous DBPs for possible enrichment in noncoding regions such as repetitive element class and families. Overall, this strategy is limited in cellular diversity, but rich in relative information of binding events at a given promoter.

By investigating these properties for 195 DBPs in K562 cells, we were able to reproduce known findings from independent data sets. For example, the number of binding events at a promoter correlates with RNA expression output (both nascent and mature transcripts) [17,18]. We also made several new observations. Specifically, we identify 1,362 promoters that do not produce a mature transcript despite having up to 111 DBP binding events. Surprisingly these promoters do not overlap with super-enhancer annotations. Therefore, we termed these promoters "reservoirs" because these promoters serve as 'reservoirs' for DBPs. While the function of reservoirs remains undetermined, they are clear deviations from the observed and previously published phenomena of more binding events correlating with more expression. In the case of reservoirs, we see a maximum number of binding events with the opposite pattern of not being expressed at all. Importantly, our reservoir annotations are distinct from super-enhancers and highly overrepresented by long noncoding RNA (lncRNA) promoters. We also observed that the human specific SVA repeat is one of the few repeat families that had specific DBP enrichment, with a total of three DBPs specific to SVA repeats. Looking further we found that SVA repeats reside adjacent to or within enhancers and are often transcribed; suggesting they may have been co-opted in late primates as enhancer elements.

Overall, we demonstrate the utility of implementing data-science and reproducibility standards to gain new insights combinations of genome-wide DNA binding events. We further note that the design of this study was intended for didactic purposes and carried out by students in a classroom setting.

## Results

We first set out to survey the encode portal for the largest number of ChIP-seq experiments that satisfied the following criterion: (i) target was considered a DNA binding protein (DBP), the experiment used validated antibodies, sequencing was performed with 100bp paired end reads and were in the same cell setting. We found the maximum number of samples that meet these requirements were performed in K562 cells (retrieval date Jan. 2020). Specifically, there are 1,076 FASTQ files comprised of 195 DBPs meeting these criteria in K562. Rather than analyzing the peaks already called by ENCODE for these experiments we chose to re-analyze the raw data using a community-curated pipeline developed by "nf-core" [20]. This approach meets the highest data reproducibility standards by using a container for all software and producing extensive documentation at every stage of analysis within the nf-core/chipseq (v1.1.0) pipeline (Fig 1A).

The nf-core/chipseq pipeline consists of documented analyses and quality control metrics that results in significant windows or peaks of DNA binding events for each replicate [20]. After the standardized pipeline gave us peak calls, we used this data to support our analysis and exploration of the data. Our approach was to use R and Rmarkdown to document the analyses. Compiling the 11 Rmarkdown files provided in the GitHub repository (https://github.com/boulderrinnlab/CLASS_2020) will reproduce all the results and figures of this study.

After calling significant peaks (MACS broad peak) for each replicate ChIP experiment for each of the 195 DBPs, we wanted to develop consensus peaks across replicates. Briefly, we filtered to peaks on canonical chromosomes and required that peaks overlap by at least 1nt in all replicates for a given DBP. Peaks that overlap in all replicates are then merged by the union of peak widths (Fig 1B and 1C). We observed five DBPs that did not have any peaks overlapping across replicates perhaps suggesting that these are promiscuous antibodies, or these proteins have heterogeneous binding across K562 cell populations (MCM2, MCM5, MCM7, NR3C1, TRIM25).

We next plotted the distribution of the number of consensus peaks for each DBP and found that many DBPs had very few peaks. In order to capture the majority of DBPs and still provide a reasonable number of peaks for analyses (e.g., permutation analyses). To determine an empirical cutoff, we chose to eliminate consensus peak profiles that were in the bottom 15$^{th}$ percentile based on the distribution of peaks across all DBPs (S1A Fig). A cutoff at the 15$^{th}$ percentile of the data resulted in DBPs that had at least 250 peaks to analyze. Overall, this results in 161 proteins to carry forward in the analysis and in losing the following proteins: ARNT BCLAF1 COPS2 CSDE1 DNMT1 eGFP-ETS2 FOXA1 KAT8 KDM4B MCM2 MCM5 MCM7 NCOA1 NCOA2 NCOA4 NR0B1 NR3C1 NUFIP1 PYGO2 THRA TRIM25 TRIP13 XRCC3 YBX1 YBX3 ZBTB8A ZC3H8 ZNF318 ZNF830.

### Promoter centric binding properties of 161 DNA binding proteins

We next plotted the relationship between the number of consensus peaks observed for each DBP and how many promoters overlapped (36,814 lncRNA and mRNA promoters). We observe a linear relationship (slope = 0.31 for mRNA and lncRNA promoters) between the number of peaks and or size of peaks and the number of overlaps with promoter regions (Fig 2A). Somewhat surprising was this trend was even more pronounced when comparing overlaps within gene-bodies rather than promoter regions (Fig 2B). This suggests we could have an observation bias at promoters where promoter binding simply increases with the number of peaks observed for a given DBP and not due to preferential binding at promoters.

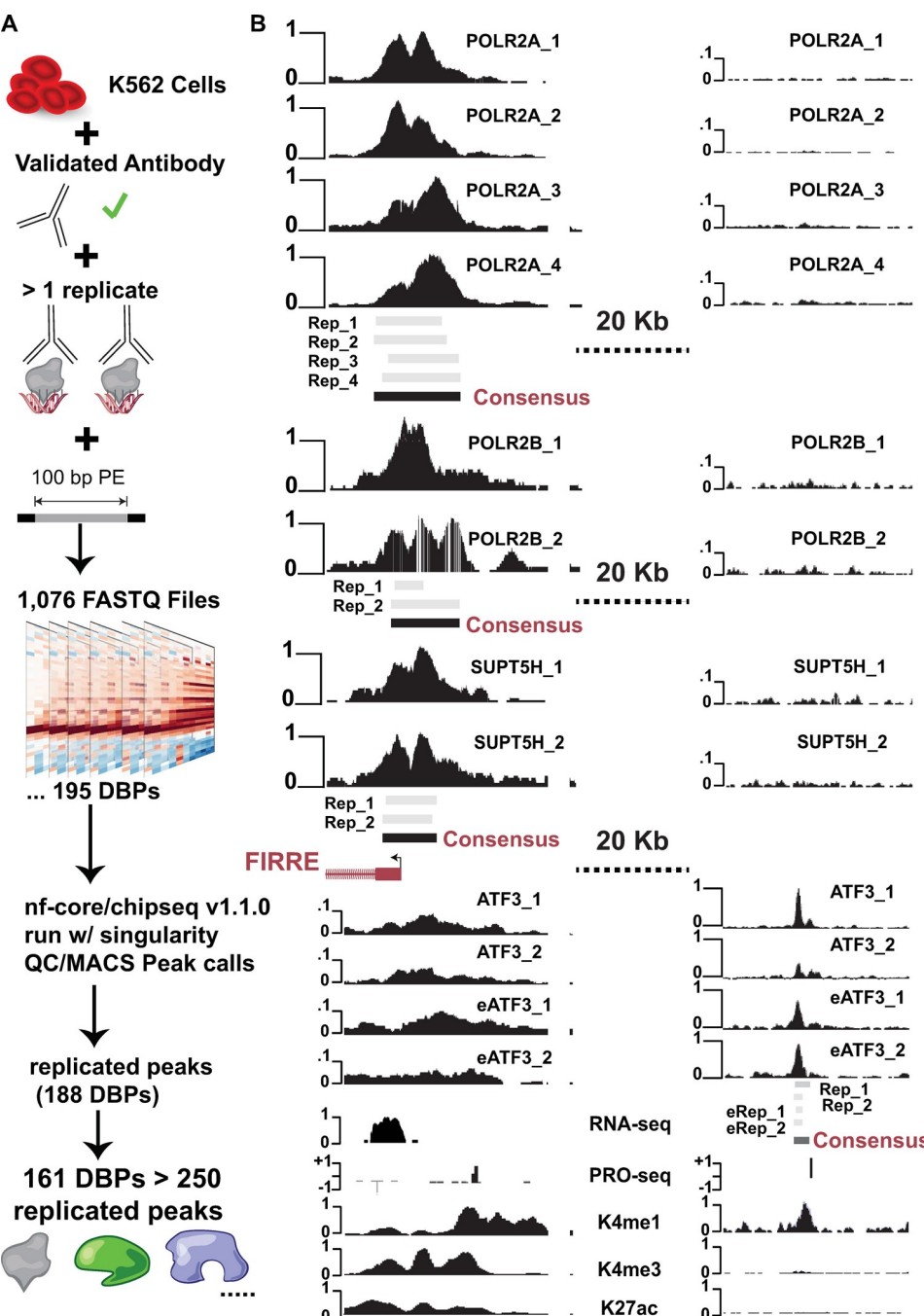

**Fig 1. Framework of ChIP-seq analyses and peak calling across replicates.** (A) Schematic of data quality requirements (validated antibody, two or more replicates, 100bp reads) resulting in 1,076 FASTQ files representing 195 unique DNA binding proteins. FASTQs were processed using the nf-core/chipseq pipeline (QC and peak calling). All FASTQ files passed nf-core quality control metrics. (B) Browser view of raw data, individual replicate peak calls and our consensus peaks. All scales are from 0 to 1 representing minimum and maximum reads in that window using UCSC auto-scale. Peaks from individual replicates are in gray and consensus peaks called are represented by black boxes.

To detect preferential binding at promoters, we took a permutation-based approach for each DBP's peak-profile across the genome. Briefly, we took the consensus peaks for each DBP

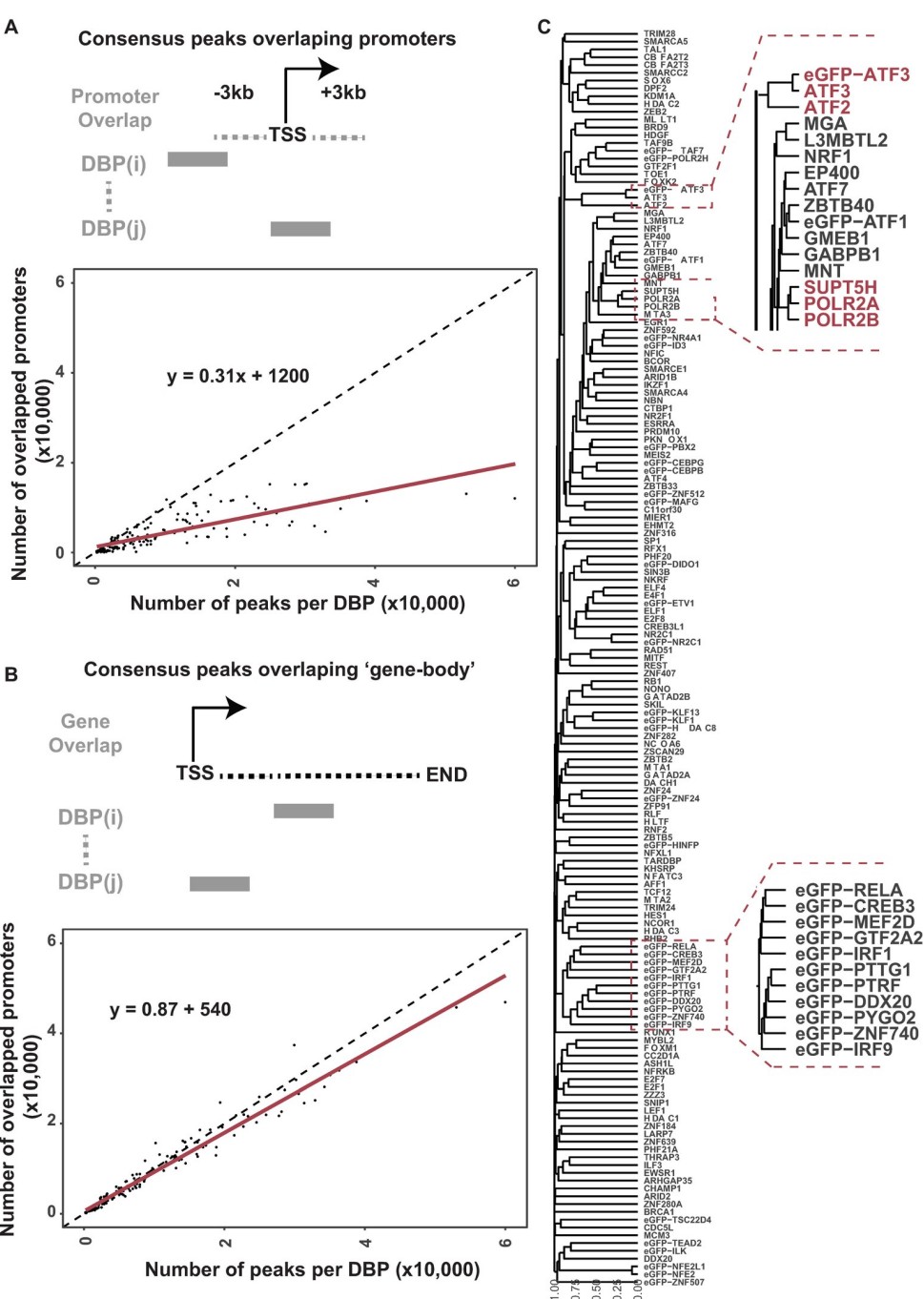

**Fig 2. Promoter binding properties of 161 DBPs.** (A) Schematic of promoter overlap strategy. Number of overlapping promoters (y-axis) per number of peaks for each DBP (x-axis). (B) Same as in (A) but for overlapping gene bodies instead of promoters. (C) Hierarchical clustering of 161 DBPs based on promoter binding profiles (consensus peaks) with zoom in on specific regions.

and randomly placed them across the genome, while controlling for (i) number of peaks, (ii) width of peaks and (iii) number of peaks on each chromosome. We then performed a Fisher exact test of the observed binding at lncRNA and mRNA promoters versus expected binding in the empirically derived null distributions. We observed that nearly all DBPs exhibit

significant overlap with promoters versus the rest of the genome, despite being involved in many different DNA regulatory processes (S1B Fig).

To more closely examine the results of our consensus peak strategy we performed manual inspection of samples with two or more replicates (Fig 1B and 1C). We find that our peaks are consistent with what would be expected of highly reproducible binding events. We see that most Pol II and ATF3 peaks show good agreement between replicates. Interestingly in this example ATF3 is not localized to the promoter but in an upstream region that could be a new-found enhancer or upstream regulatory element. Overall, these analyses are consistent with our consensus overlap strategy representing expected and newfound features in peak size profiles.

## Global analysis of similarities in binding profiles

To determine if there were underlying similarities and differences of 161 DBPs that passed our conservative filtering, we first performed hierarchical clustering (Fig 2C) on binary vectors representing binding events on 36,814 lncRNA and mRNA promoters defined in GENCODE 32 where 1 = bound, 0 = not bound for each promoter and DBP. As a quality control check, we looked for clustering of known factors. The binary vector profiles validated that POLR2A, POLR2B, and SUPT5H form a distinct cluster. Known family members, such as ATF3 and ATF2 co-cluster together as well, along with the eGFP-ATF3 control. This indicates that these DBPs had similar binding profiles with or without the eGFP tag. However, 11 cases of eGFP-tagged samples clustered together, despite having widely different functions. This may suggest that in some rare cases the tag can alter the binding profiles in a manner that is more consistent with the tag than DBP function.

As an unbiased approach to find underlying properties in DBP binding profiles, we also performed UMAP [21] dimensionality reduction for the global binding profile of each DBP (Fig 3A). Briefly, UMAP uses algebraic topology to reduce the data dimensionality. We further clustered this reduced representation using density-based clustering (HDBscan [22]). We observed a total of seven clusters. Similar to binary clustering, we identify a clear cluster of POLR2A, POLR2B, and SUPT5H and other basal transcriptional associated factors (TAF) as would be expected. This is another example of high reproducibility as POL II has three different antibodies with two replicates each that are all highly concordant with thousands of peaks each.

Next, we compared specific features of the DBP with their position in the reduced space by mapping metadata (e.g., type of DNA binding domain) onto the UMAP points (Figs 2A and S2A–S2D). We found no clear association with (i) type of DNA binding domain, or (ii) annotation as a transcription factor, (iii) RNA-seq expression of bound genes, or other properties. Collectively, these results recapitulate known biological functions of DBPs while including potential new factors across these different promoter regulatory functions.

## Promoter binding specificity of 161 DNA binding proteins

We next wanted to assess the underlying promoter features associated with each DBP. Specifically, we wanted to determine where each DBP is bound relative to the TSS of 36,814 lncRNA and mRNA promoters. To this end, we generated 'binding profile plots' by calculating the read counts across all promoters centered at the TSS with 3kb flanking up- and down-stream (Figs 3B and S2E and S2F). We next clustered the 161 DBPs based on their promoter profile plot. We split the dendrogram into clusters by 'cut-height' (h = 65). We observed 4 distinct clusters with at least two DBPs. The first distinction is that about half exhibit a narrow peak profile (71) and half with a broader peak profile (74). In both cases these profiles peak near the TSS.

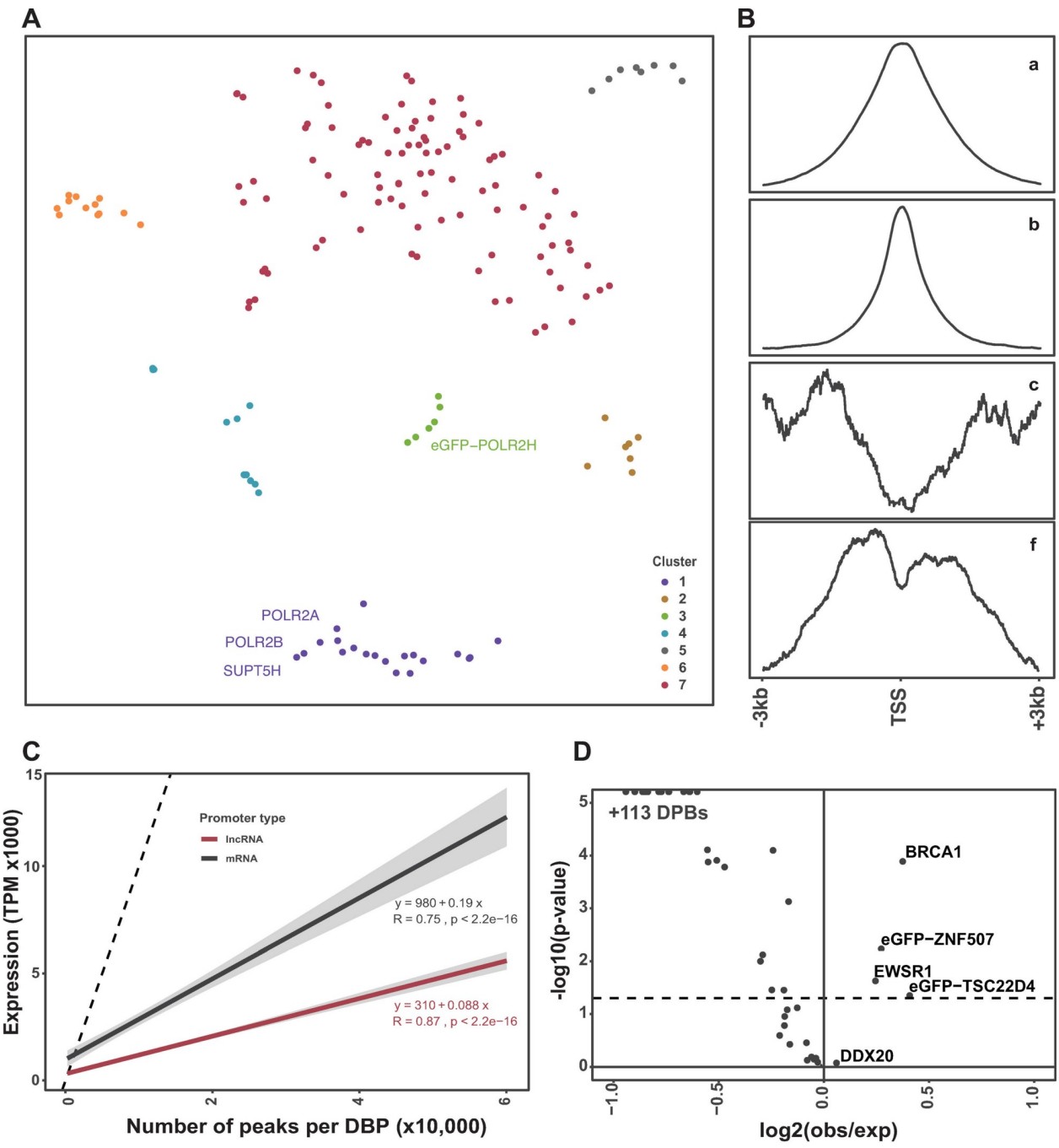

**Fig 3. Binding properties of DBPs and expression output of promoters.** (A) UMAP dimensionality reduction to identify DBPs with similar promoter binding profiles. (B) Four distinct clusters of binding patterns around promoter TSSs with 3Kb up and downstream. Line is the average profile of all peaks in each cluster. (C) The number of peaks per DBP versus number of mRNA or lncRNA promoter overlaps. X-axis is the number of DBPs overlapping either lncRNA (red) or mRNA (black) promoters. (D) Chi-squared test for enrichment of DBPs between lncRNAs and mRNAs. The x-axis is the log2(observed over expected) and y-axis is the -log10(p-value).

Interestingly, 6 genes (eGFP-PTRF, ZBTB33, SMARCA5, HDGF, eGFP-ZNF512, eGFP-ZNF740) have the inverse pattern: depletion of binding at the TSS with strong enrichment at flanking regions (S2E and S2F Fig).

Previous studies have identified several differences in binding features at coding (mRNA) compared to noncoding (lncRNA) promoters. Here we wanted to independently test this across 161 DBPs to determine if there was an enrichment or depletion at mRNA versus lncRNA promoters. We counted the number of lncRNA and mRNA promoter overlaps separately and observed the same linear trend of more peaks resulting in more binding events for both lncRNAs and mRNAs. However, the slope for mRNA is 0.19 (R = 0.75, P < 1e-10) and lncRNA is 0.088 (R = .87, P < 1e-10) suggesting a two-fold reduction on an average lncRNA promoter (Fig 3C).

We then performed permutation analysis (above) for lncRNAs and mRNAs to determine if the observed overlap is greater than expected by chance (S1B Fig). Similar to our previous observation, nearly all DBPs were significantly (Fisher-exact P < 0.05) enriched at both lncRNA and mRNA promoters yet with a smaller magnitude of enrichment of binding events on lncRNA promoters (similarly to previously reported [17]). We observed two DBPs that were significantly depleted: BRCA1 on mRNA promoters, and ZNF507 on both lncRNA and mRNA promoters. Four DBPs showed neither enrichment or depletion at lncRNA or mRNA promoters. In total 155 of the 161 tested DBPs were enriched at lncRNA and mRNA promoters more than expected by chance (S2G Fig).

To more comprehensively analyze binding preferences of each DBP we used the output from the nf-core/chipseq pipeline produced by HOMER [23]. Specifically, HOMER annotates peaks with their overlapping genomic features: TSS, intronic, exonic, transcriptional termination sites, promoter, and intergenic (S3 Fig and S3 Table). Consistent, with the above findings a vast majority of the DBP binding occurs between the promoter and gene-body. Interestingly, BRCA1 and ZNF507 are exceptions that have very little promoter binding and cluster together as would be expected from the permutation analyses above (S2G Fig).

Our previous permutation test above demonstrated that most DBPs bind both lncRNA and mRNA promoters more than expected by chance. But this approach does not account for DBPs that may prefer lncRNA or mRNA promoters. Thus, we hypothesized that some DBPs may have a bias in binding for mRNA relative to lncRNA promoters and vice-versa. To test this, we performed a Chi-squared test to compare the number of binding events for each DBP at lncRNA versus mRNA promoters. Interestingly, although most DBPs are enriched on mRNA promoters, there were a few with a relative bias toward lncRNA promoters (P < 0.05): BRCA1, eGFP-ZNF507, EWSR1, eGFP-TSC22D4 (Fig 3D). Interestingly, BRCA1 prefers to bind outside of promoters, yet if it does bind a promoter BRCA1 prefers lncRNA over mRNA promoters.

## Repeat family and class binding preferences for 161 DNA binding proteins

In order to determine if DBPs are enriched or depleted in TE classes and families we performed a permutation enrichment analysis. As above, we randomly shuffled peaks around the genome and calculated the number of overlaps with repeat family and classes from RepeatMasker Open-3.0 occurring by chance (Fig 4A).

We observe that some classes, such as Simple Repeats and tRNAs, were enriched for most DBPs, while others, such as the LINEs and Satellites, were depleted for most DBPs (Fig 4A). The LINE class was depleted of all DBPs with the exception of five DBPs with zinc finger-like motifs (ZNF507, ZNF316, ZNF184, ZNF24 and ZNF512). Additionally, the LTR class was depleted for most DBPs, but enriched for a subset of 23 DBPs (Fig 4B).

Overall, we found that most small TE families were not significantly enriched or depleted for specific DBPs. However, a subset of 23 DBPs were enriched in the ERV1 family, but depleted in the L1 family. These 23 DBPs are the same that were enriched in the LTR class.

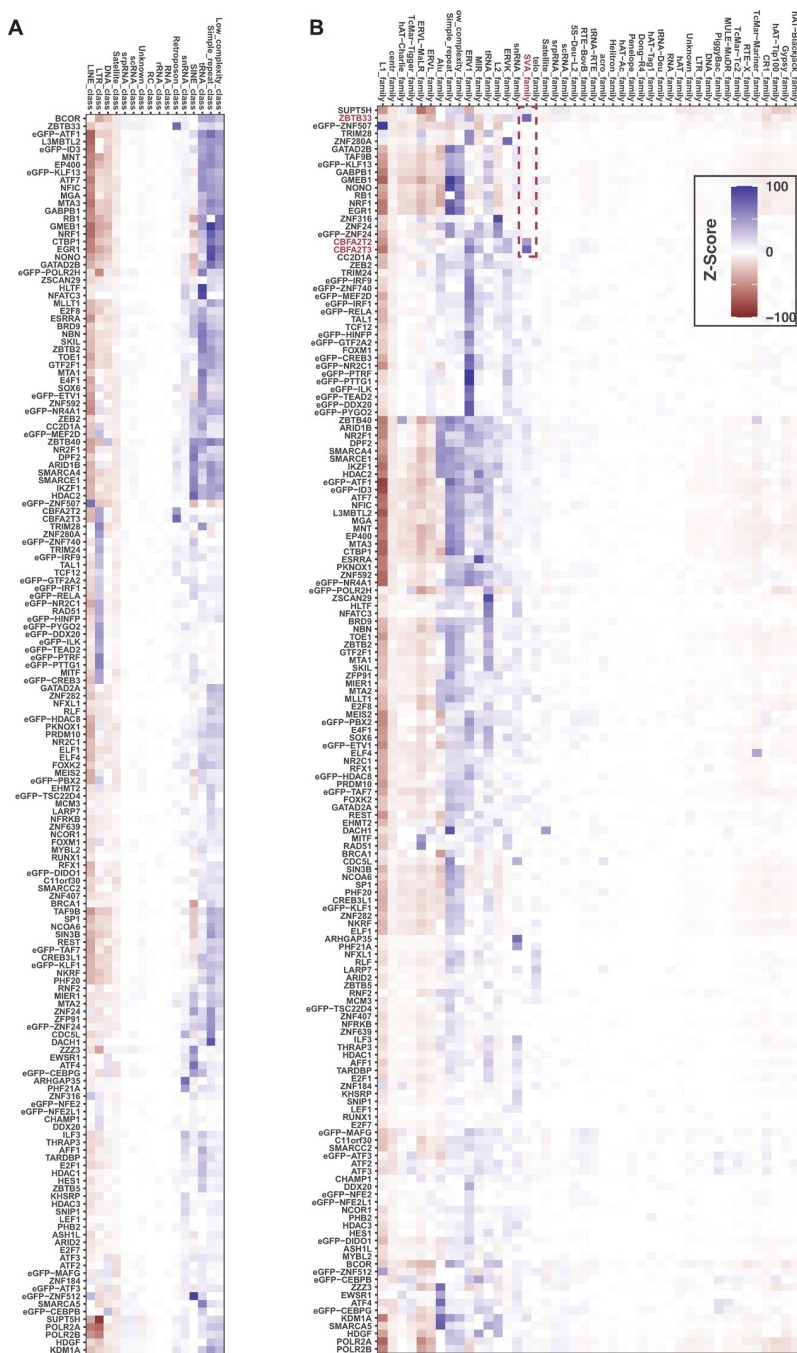

**Fig 4. Many DBPs are enriched or depleted on repeat classes and families.** (A) Heat map of Z-scores of observed overlaps of each DBP versus the overlap distribution of 1,000 random permutations of each DBPs profile genome wide. Red indicates depletion and blue enrichment (negative versus positive Z-scores respectively). The observed and permuted Z-scores are for overlaps with repeat classes. (B) The same permutation analysis as in (A), but for observed versus permuted overlaps with repeat families. Red indicates depletion and blue enrichment.

This is consistent with ERV1 family TEs being a part of the LTR class. Similarly, the MIR family shows a similar enrichment pattern to the tRNA family (Fig 4B). Thus, using this approach we can provide a map of which DBPs are specifically bound to which repeat family.

We did observe a subset of 6 DBPs (NUFIP1, ZC3H8, PHF21A, ARHGAP35, NCOA4, PYGO2) enriched in snRNAs, but no other TE family. Each of these DBPs, except NCOA4, contains a zinc-finger-like DNA binding domain, and a few (NUFIP1 and ZC3H8) are known to be a part of the snRNA biogenesis pathway, perhaps suggesting some form of feedback. The L1 family is depleted for almost every DBP, but is highly enriched for ZNF507, an interaction which has been previously described in an undergraduate thesis and confirmed genome-wide here (https://web.wpi.edu/Pubs/E-project/Available/E-project-042618-111020/unrestricted/MQP.pdf).

## The human specific SVA repeat family has enhancer like features

Although most families are not enriched for specific DBPs, the human specific SVA repeat family is specifically enriched for three DBPs: ZBTB33, CBFA2T2, and CBFA2T3. Interestingly, all three of these DBPs are known transcriptional repressors. The SVA family is the youngest family of TEs, is enriched in gene-rich areas of the genome [24–26], and can cause human disease [25]. Based on these interesting features we further explored the binding of these factors on the SVA repeat.

We first retrieved histone modification ChIP-seq data for K562 cells from ENCODE and visualized the coverage centered on the 5,882 SVA repeats with 5kb up- and down-stream. We find that H3 Lysine 4 mono-methylation (H3K4me1) is the only histone modification enriched on SVA elements–all others were depleted (S4A Fig). Moreover, the enrichment of H3K4me1 is on the 5' end of the SVA element suggesting it could be an insulator for enhancers or part of the enhancer element. This pattern is so sharp we were concerned about mappability to the SVA element–despite observing the 5' enrichment of H3K4me1. We reasoned that ZBTB33, CBFA2T2 and CBFA3T3 should be enriched across the SVA element. We performed the same analysis above for the these 3 DBPs and find there is strong mapping to these SVA regions. (S4B Fig). We next looked at the expression level of SVA elements relative to other repeat family members. Interestingly, we observed that SVA elements have more transcription (S4C Fig) than LTR family members that are known to function as promoters [27]. Together, these results demonstrate that the SVA region has enriched and fully mappable coverage of H3K4me1, ZBTB33, CBFA2T2, CBFA3T3 and are expressed.

Of the 5,882 SVA elements genome-wide, 255 SVAs were found to contain consensus peaks for all three enriched DBPs: ZBTB33, CBFA2T2, CBFA3T3. We took the same approach above for this subset of bound SVA elements. We see even stronger enrichment of H3K4me1 (Fig 5A) and also coverage by ZBTB33, CBFA2T2 and CBFA3T3 (Fig 5B). Interestingly, the shape and position of ZBTB33 is distinctly different than that of CBFA2T2/3T3 (Fig 5B). It suggests that ZBT33 binds on the 5' region near H3K4me1 and CBFA2T2/3T3 have overlapping positions on the 3' end of SVA elements. Closer examination of nascent, steady state RNA-sequencing (see below) and H3K4me1 ChIP shows a very interesting pattern of the SVA elements being transcribed and or producing bi-directional RNAs in H3K4me1 enriched (Fig 5C and 5D). This is very similar to what has been seen for enhancer regions genome wide [28,29]. Thus, the SVA transposon may have evolved (neutrally or positively) to 'co-opt' binding of DBPs adjacent or within enhancer regions.

## Promoter binding of 161 DNA binding proteins versus promoter expression output

Here we set out to investigate how binding events at individual promoters relate to the concomitant expression of the gene-product at that promoter. To this end, we analyzed ENCODE K562 total RNA sequencing data from two replicates. We calculated the average read coverage

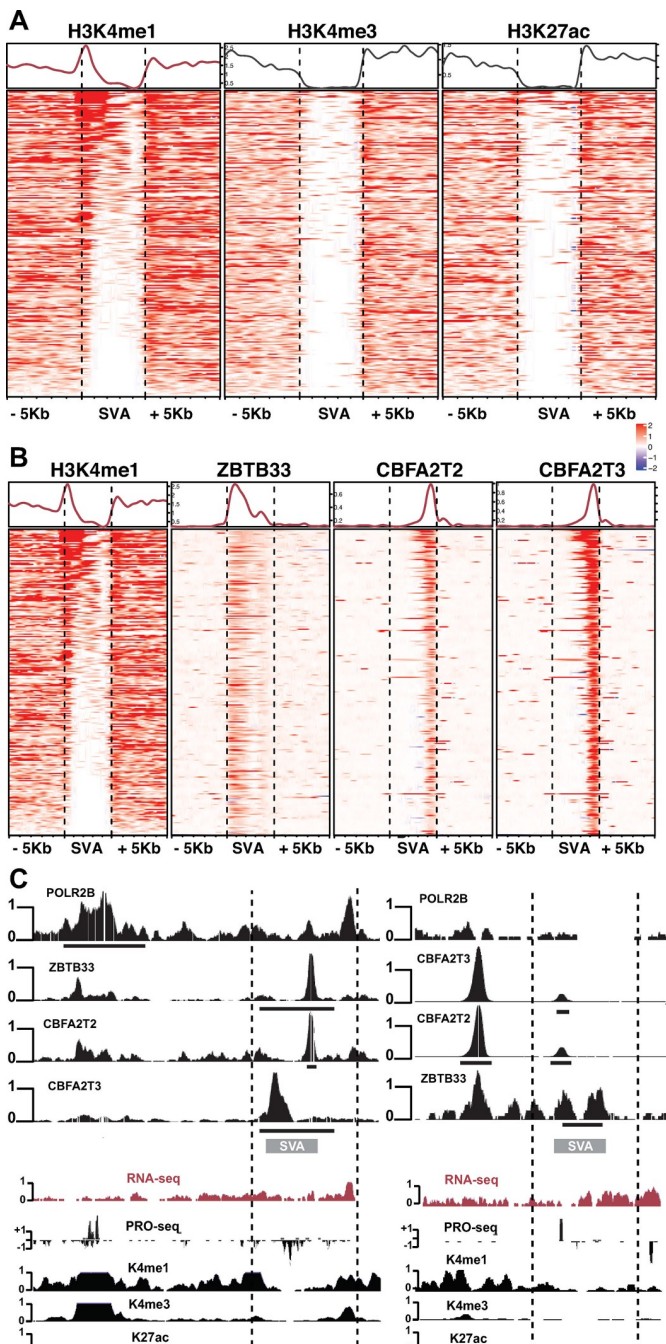

**Fig 5. Human SVA repeats are enriched for DBPs and enhancer properties.** (A) Heatmap of histone modification reads centered on SVA and 5Kb up- and down-stream for the 255 SVA elements containing ZBTB33 and CBFA2T2/ T3 peaks. Here red indicates enrichment, while blue indicates depletion. Above is the average profile line of enrichment within and outside SVA elements. (B) Same as (A) but coverage of ZBTB33 and CBFA2T2/3T3. The H3K4me1 plot is same as in (A) for direct comparison. (C) Browser examples in the same format as Fig 1.

across replicates and quantified by transcripts per million reads (TPM); while considering the variance between replicates in further analyses. We first asked if the number of binding events at a promoter correlated with expression. We observed a positive correlation (R = 0.6,

P < 2.2e-16) between the number of DBPs bound at a promoter and expression output of the promoter (S5A Fig).

We next wanted to determine if this trend is similar for mRNA and lncRNA promoters separately. Indeed, we see that both lncRNA and mRNA promoters have a positive correlation to binding events and expression output (Fig 6A). We observed that lncRNAs have lower expression in general than mRNA as previously determined [30–33]. Yet despite these expression differences, both exhibit a positive relationship between number of binding events and promoter activity. This is consistent with observations in a previous study using a different yet overlapping subset of 73 DBP ChIP datasets [33].

Although we saw a linear trend with binding events and expression output above, we wanted to refine this analysis to a binned approach. Specifically, we binned lncRNA and mRNA promoters by expression output. Based on the distributions of TPMs calculated for all genes we made cutoffs approximating quartile bins for "Off", "Low", "Medium" and "High" expression (Off: < 0.001 TPM ($0^{th}$-~$25^{th}$ percentile), Low: (0.001,0.137] TPM (~$25^{th}$-$50^{th}$ percentile), Medium: (0.137,3] TPM (~$50^{th}$-$75^{th}$ percentile), and High: >3 TPM (~$75^{th}$-$100^{th}$ percentile). These are simply relativistic bins to compare different levels of expression represented in the linear relationship between binding events and expression. Interestingly, at 'low' and 'off' expressed promoters there is no difference in binding event distributions between lncRNA and mRNA promoters (Fig 6B). Thus, they both have similar numbers of binding events—and can have dozens of DBPs bound—despite having little to no expression output. In contrast, mRNA promoters show significant increases in binding events, compared to lncRNA promoters, at medium and high expressed promoters. Thus, in the middle to high ranges of expression is where we begin to see the differences between mRNA and lncRNA promoters. Collectively, these results identify over a thousand promoters that resemble the DBP content of highly-expressed promoters yet do not have any detectable expression by RNA-seq.

## Promoters with numerous binding events but lack gene-expression output

Although the vast majority of promoters demonstrated increased expression with more DBP bound at their promoter, there is a clear subset of promoters that deviates from this trend. Specifically, we observed over a thousand promoters that have numerous DBPs bound, but do not produce a transcript identified by RNA-seq; in contrast to all other promoters tested that have very high expression with more DBPs bound. Based on these promoters that deviate from common promoter regulation we wanted to further characterize the global properties of this subset of promoters. First, we made density plots of the number of binding events at promoters. We observed a bimodal distribution of binding events where the cutoff between the distributions is around seven binding events at a promoter (54% percentile). Based on these two distinct distributions, we focus our analysis on those promoters with more than seven binding events (S5A and S5B Fig) and further required that the RNA-seq output was less than 0.001 TPM. This resulted in 1,362 promoters which had a relatively high number of binding events but lack of RNA-seq output from these promoters. Interestingly, 981 of the 1,362 are comprised of lncRNA promoters (S5C Fig). This is a significant over-representation of lncRNAs in these high-binding non-expressed promoters over what would be expected by chance (hypergeometric p-value = 1.1 x $10^{-88}$).

There are two trivial explanations that could explain these high binding low expression promoters: (i) these are simply super-enhancer [34–36] annotations (as they share similar properties of many binding events) and or (ii) the promoter is regulating a neighboring gene.

Our first concern is that super-enhancers (SE) share the similar property of many binding events, we wanted to determine how many of these regions were super-enhancers. For super-

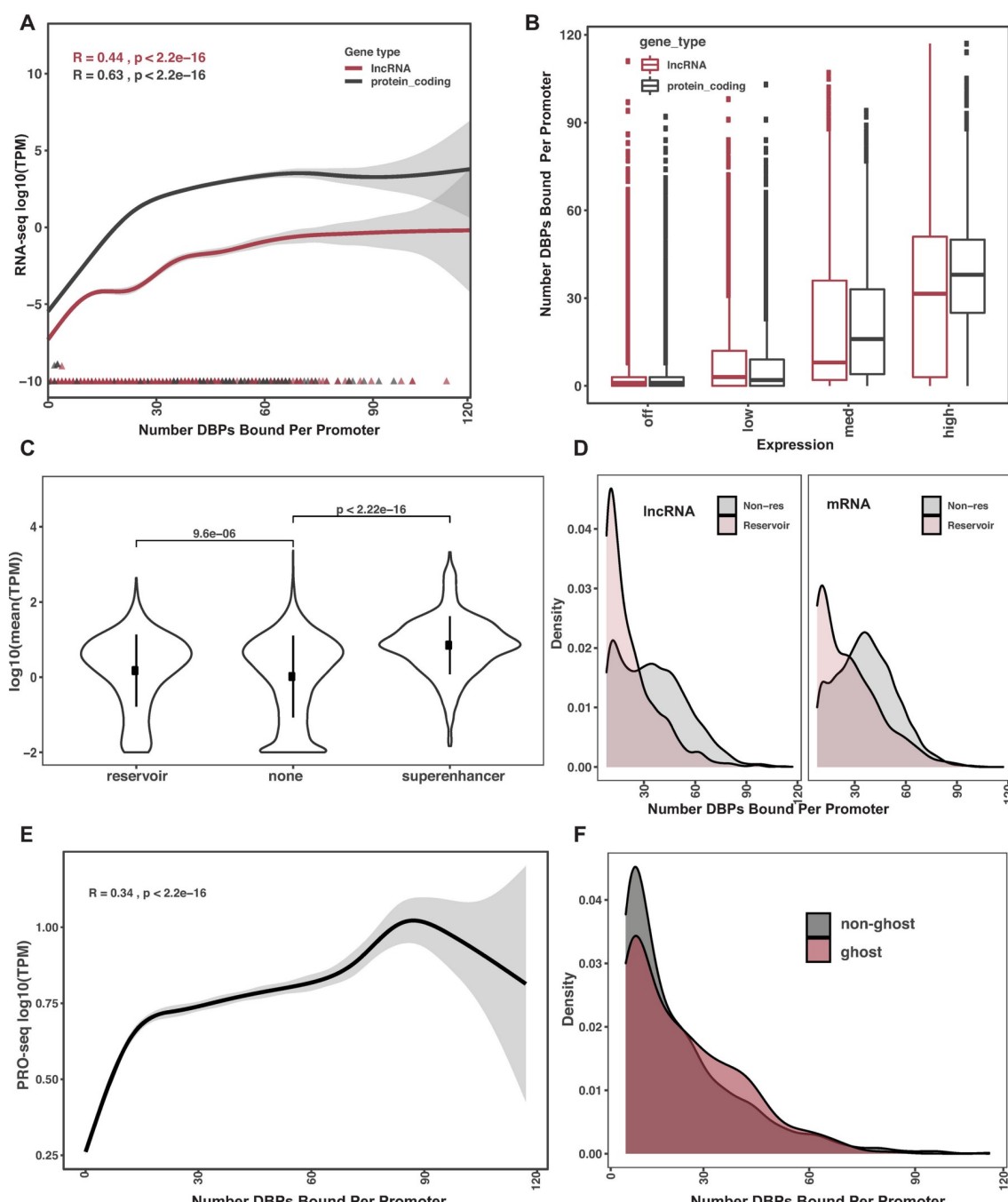

**Fig 6. Reservoir promoters are comprised of ghosts and zombies.** (A) Number of DBPs bound to a promoter (x-axis) versus $\log_{10}$(TPM) of transcription as measured by total RNA-seq. (B) Box plot comparing mRNA (black) and lncRNA (red) expression as a function of off, low, medium, and high expressed transcripts. Y-axis is the number of DBPs bound and X-axis is each category. (C) Y-axis is the mean expression level in windows of five genes excluding the center gene. X-axis is by category of windows containing a reservoir, non-reservoir or super-enhancer. (D) Density plot of number of DBPs bound at a promoter at expressed (grey) versus non-expressed promoters (red), separated by lncRNA and mRNA promoter types. (E) Nascent TPM expression within the 6kb promoter region (Y-axis) compared to number of DBPs bound at a promoter. (F) Density plots of DBPs ghost reservoirs (those without nascent expression, PRO-seq TPM < 0.001) vs those with detectable nascent expression (zombie reservoirs).

enhancer annotations we used the SE-DB [37] that is comprised of 331,601 super-enhancers from 542 tissues and cells, including K562. We first retrieved the SE annotations in K562 with

the hg19 reference genome alignments. We then lifted over these annotations from hg19 (732 annotated SEs) to hg38. We found 714 annotations have one match to the genome and took a conservative approach of not including the 18 SEs with multi-mapping in the genome (often too many chromosomes). Of these 714 regions, 35 overlapped with the 1,363 reservoirs (P = .991 Hypergeometric). Thus, reservoirs are distinct from SE annotations and are enriched with repressor complexes unlike SEs.

Another concern is that these promoters we identified could regulate a neighboring gene; this would be most obvious for bidirectional promoters. Thus, we first defined a set of promoter types: (i) bidirectional, if another promoter on opposite strand overlaps within 1,000 bp upstream of the TSS on the opposite strand (147 / 11%); (ii) multiple nearby promoters, if there is more than one promoter on either strand within 1,000 bp (91 / 7%); (iii) nearby on same strand if there is another promoter upstream within 1,000bp (113 / 8%); (iv) none (1,011 / 74%), if there are no promoters within 1,000 bp (S5D Fig). Collectively, very few reservoirs had shared promoters of any type i-iii (26%), thus this cannot likely account for the lack of transcription at the observed or neighboring promoter (since there are so few). Nonetheless we calculated the TPM of promoter(s) neighboring reservoirs. We observed that 68% of these shared promoters did have a neighboring gene expressed (subcategories in S5E Fig) for a total 240 (15%) of reservoirs that could affect neighboring gene expression. Thus, neighboring promoters of any orientation cannot account for the general lack of expression observed at reservoirs (Chi-squared p-value 2e-22).

Although bidirectional expression cannot explain why these promoters seem inert, we wanted to look more globally at the transcription environment of these promoter regions and their 5 neighboring genes. Specifically, we used a "sliding-window" approach to calculate the median TPM expression value for windows of 5 genes. Each window is centered on one gene and the mean of the neighboring four genes is calculated excluding the center gene. We first plotted the distribution of windows where the center gene is a reservoir compared to those with non-reservoir center genes. We also removed the 35 reservoirs that were annotated as super-enhancers. We observed that the Wilcoxon test statistic (Fig 6C) between means was significant ($P < 9e-06$), however the means were very similar (mean = 7.2 for reservoir, mean = 8.4 for non-reservoir). To be sure this is not an artifact of our permutation analysis we performed the same analysis for windows of genes centered on super-enhancers versus non-super-enhancers. Indeed, we see that super-enhancers reside in regions of significantly higher transcriptional activity ($p < 2.5e-12$) with a large fold change (4.5x) in mean expression (mean super-enhancer = 37 TPM, mean = non-super-enhancer 8.4 TPM) (Fig 6C).

Finally, we wanted to define a conservative or stringent subset of reservoirs that lack expression in the 6Kb window from the TSS of a reservoir (3Kb up- and downstream from the TSS); not just lacking expression in the gene annotation associated with the reservoir promoter. To control for variation from lab to lab in k562 RNA-seq we added three RNA seq data sets from three different labs that were selected for poly-adenylation (PolyA+ seq). Next, we quantitated total and PolyA+ RNA-seq from k562 cells in a 6Kb window surround the TSS of all reservoir promoters. We observed 356 reservoirs did not have expression within 6Kb of the TSS of a defined reservoir. Although we cannot determine if the reservoir is regulating a neighboring gene within 3Kb of the reservoir, we include these as the most conservative set of reservoirs (S2 Table). Overall, we define reservoirs as promoters with multiple binding sites and lack of expression downstream of the reservoir promoter.

Collectively, these results identify a subset of promoters that appear to be a 'holding place' for DNA binding events. Thus, we will refer to these promoters as 'reservoirs' since they: (i) are distinct from super-enhancer annotations; (ii) are located in more transcriptionally silenced neighborhoods; (iii) share the property of many DNA binding properties as those

promoters that are highly expressed and (iv) have no expression output as measured by RNA sequencing. Thus, reservoirs are simply a small subset of promoters that deviate from global trends of promoter properties–with no known functional role(s).

## DNA binding properties of reservoir promoters

To understand if reservoir promoters are enriched for certain DBPs, we compared the density of DNA binding events at lncRNA and mRNA reservoir and non-reservoir promoters which had greater than seven binding events. We observed a shift toward fewer binding events for both lncRNA and mRNA reservoirs (Fig 6D). However, it's notable that there are still reservoirs along the whole range of DBP binding. Finally, we wanted to determine if DBP binding profiles are similar or different at reservoir promoters to non-reservoir promoters. To this end we generated a promoter-profile plot for each DBP that had sufficient binding in reservoirs (>100 peaks) directly overlaying the binding of reservoir and non-reservoir promoters. We observed that binding trends are similar across both types of promoters, but for many DBPs the binding is more diffuse around the TSS. (S6 Fig).

Next, we wanted to determine if there was enrichment of certain DBPs on reservoirs. Using a Chi-squared test to compare the number of bound promoters for reservoirs versus non-reservoirs we observed that 31 DBPs were depleted on reservoirs and only one gene enriched (P < 0.001 and > 2-fold depletion/enrichment, S5F Fig). This is in contrast to the lncRNA and mRNA comparisons above where we saw global depletion of all DBPs on lncRNA promoters (Fig 3D). Thus far, reservoirs are deviant from all trends observed for the other ~33,000 promoters tested above.

We wanted to further globally characterize reservoir promoters using UMAP dimensionality reduction as in Fig 2A. Unlike with all promoters we only observe two distinct clusters across reservoirs (S5G Fig). However, gene-ontology analysis revealed that both clusters are strongly enriched for similar processes such as regulation of transcription (P < 1e-20). Perhaps as expected, Pol II and associated transcriptional machinery are some of the most significantly depleted from reservoirs, consistent with their lack of expression. Despite a global depletion of Pol II at reservoirs, we were surprised that over a quarter of reservoirs (417) had Pol II binding events, suggestive of 'paused' transcription. While only one DBP (eGFP-TSC22D4) reached the fold-change threshold, two more were found to be significant (P < 0.001) with small enrichments. All three are associated with repressive activity. TSC22D4 and CBFA2T2 are both known repressors while EHMT2 facilitates transcription repression through methylation of H3K9. Collectively, these findings show that reservoir promoters are distinct from super enhancers, bound by many DBPs and yet are not transcribed.

## Nascent expression and chromatin properties of reservoir promoter

Since reservoirs don't have mature transcriptional products despite many promoter binding events, we next examined if reservoirs have "nascent" transcription detected via PRO-seq (reviewed[34]). These approaches are so precise they can identify specific DBP binding sites through PRO-seq nascent RNA read out [38,39]. Thus, we hypothesized that reservoir promoters would exhibit nascent transcription owing to so many DNA binding events. This could also be similar to more well established "paused" promoters as reviewed [40].

To determine the nascent transcription properties of reservoirs, we obtained two replicate pro-seq data sets that measure the amount of nascent transcription at a promoter. We used "Rsubread" [41] to calculate TPM values of nascent transcription across the same 6 Kb promoter window defined for DBP binding. We first plotted the relationship of nascent expression at reservoirs versus non-reservoirs (S7A Fig). Although statistically different (P < 3e-9)

the distributions are fairly similar for reservoirs (mean = 0.41) and non-reservoirs (mean = 0.51) with a fold change of only 1.25. Thus, consistent with lack of RNA-seq expression, reservoirs also have slightly lower nascent expression than non-reservoirs (S7A Fig). Next, we compared the relationship between the number of DBPs bound and nascent expression levels (Fig 6C). Similar to what was observed for RNA sequencing and previous studies [17,33] (Fig 3C), nascent transcription also has a significant (R = .3, P < 2e-16) positive correlation with the number of DBPs bound at that promoter (Fig 6C).

Interestingly, we observed a subset of reservoirs that have many DNA binding events but do not have nascent transcriptional activity. Specifically, we found 355 (25%) promoters with more than 7 and as many as 60 binding events that have neither nascent nor mature expression (PRO-seq TPM < 0.001, Fig 6F). We refer to these reservoirs without nascent or mature transcription as 'ghosts', as there is no presence of transcriptional activity. We also found 964 promoters with more than seven binding events that had no mature expression but did have nascent expression. These are referred to as 'zombies,' as there is some presence of activity.

We next investigated if the chromatin environment discriminates between ghost and zombie promoters. We therefore retrieved ENCODE ChIP data from K562 for a euchromatic and heterochromatic histone modification; Histone 3 Lysine 27 acetylation (H3K27ac) versus Histone 3 Lysine 27 trimethylation (H3K27me3) respectively. To this end, we downloaded peak files called in two independent replicates for each histone modification from ENCODE analysis pipelines. To validate our re-analysis of these ChIP-seq experiments we first determined if H3K27ac correlates and H3K27me3 anticorrelates with global nascent transcription as would be expected. Indeed, we see that those promoters containing H3K27ac have increased nascent expression (P < 2e-16, fold change = 4) (S7B Fig). Similarly, we checked the trend for H3K27me3 status (S7C Fig). As expected, we see that promoters containing H3K27me3 have lower nascent expression (P < 2e-16, Fold change = 0.3, S7C Fig).

Having validated that our analysis of PRO-seq faithfully represents known biological processes (e.g., H3K27ac enriched with higher expression) we wanted to zoom in only on reservoirs. We first compared H3K27ac status versus nascent transcription levels on reservoirs. As was seen with all promoters we see a significant difference in nascent expression between H3K27ac containing reservoirs and those without that mark (P < 0.0006, fold change = 1.65, S7D Fig). Similarly, H3K27me3 status on reservoirs is negatively associated with nascent expression levels (P < 0.0002, fold change = 0.55, S7E Fig). However, chromatin environment doesn't fully explain the presence of zombie promoters, as there are promoters with and without nascent expression in each category of chromatin state.

To understand the difference between ghosts and zombies, we compared DBP binding events, the distribution of nascent transcription, and histone marks. We did not observe a significant difference in distribution of DBPs between ghosts and zombies (P = 0.064, fold change = 1.04, Figs 6F and S7F). Thus, unlike all other cases tested, the number of DNA binding events cannot account for the difference in those that do and don't have nascent expression. Collectively, these findings demonstrate that more than 60 DBPs bound to the same promoter do not exhibit nascent nor transcript production and are 'ghosted by Pol II'. All properties identified above can be found in S2 Table.

## Discussion

A fundamental question in biology is to understand when and where DBPs localize on a given promoter and in turn how these combinations affect expression output. Thanks to heroic efforts by ENCODE and other genome consortium efforts we now have standardized DNA binding profiles for hundreds of DBPs [12–15,32]. Moreover, these datasets go through several

quality control measures before being released by ENCODE (see ENCODE portal). Thus, these important resources provide two opportunities: one for data-reproducibility standard advancements based on such well documented data; and a second to re-analyze these data-sets to find novel insights into the genome-wide localization of DNA binding proteins.

This study found a vast majority of ENCODE data to be highly reproducible—both with known biology and in data quality. However, we do note that it may be recommended to be sure replicates have reproducible peak profiles as we observed a few ChIP-seq experiments that did not have any overlapping replicate peaks. This led us to identify 5 (2%) experiments that did not have any reproducible peaks. However, a majority of the experiments (98%) have peaks that overlap in all replicates as applied in this study. Moreover, taking into account the number of observations (promoters) it is needed to be sure there are sufficient replicable peaks called for each DBP. We found 30 more samples that had fewer than 250 peaks between replicates (15th percentile). Considering the number of observations (promoters) it is also important to be sure there are sufficient peak numbers for permutation analysis and statistical comparisons. Finally, we noted that many of the proteins tagged with "eGFP" had similar binding profiles based on the tag and not DBP function (Fig 2C). We did not see differences in number or sizes of peaks compared to antibody-based ChIP. Yet it is surprising that 15 different DBPs all cluster together based on the "eGFP" tag despite diverse biological roles and all having similar consensus peak profiles.

These large and standardized data-sets also provide a unique opportunity to search for novel insights into the relationship of DBPs and expression output. Thus, we can compare 161 DBPs from the perspective of a promoter to determine how many bind and how this influences promoter output. Consistent with two recent studies using orthogonal datasets and approaches [17,18] we found that the more DBPs at a given promoter the more it tends to be expressed. This was similar for lncRNA and mRNA promoters alike. This analysis similarly validated these studies finding that mRNA promoters are more enriched in general than lncRNAs for DBPs [17,18].

Surprisingly, we observed 1,362 promoters had numerous DBPs (more than seven and up to 111 DBPs on one promoter) bound yet did not have expression output. This is in sharp contrast to the general properties observed of approximately ~35,000 other promoters that show a clear linear relationship of binding events and RNA expression output. Although this is a small subset of promoters, they represent a striking deviation with numerous DBPs bound, but lacking active transcription. In fact, these promoters had similar DBP events as the most highly expressed mRNA promoters. We termed these regions reservoirs as they seem to be a holding spot for DBPs. It is important to note that although reservoirs have very distinct promoter regulation there is no known function associated with them. Future experimental studies will be needed to determine what roles, if any, reservoirs play in genome regulation.

Notably, reservoirs are highly over-represented for lncRNA promoters relative to mRNA promoters (p < 2 e-12). We also determined that reservoirs are not super-enhancers previously defined by having many DBP binding events. Unlike super-enhancers, reservoirs have many different DBPs bound rather than many binding events of cell-specific transcription-factors in a defined region [34–37,42]. Another difference from super-enhancers is the lack of Pol II, although we do find that a quarter of reservoirs do have Pol II machinery bound. Perhaps suggesting that they are "paused promoters" [40,43] potentiated with up to 111 DBP binding events.

Further investigation into reservoirs revealed that almost half produced "nascent" transcription as measured by PRO-seq. This is consistent with the above hypothesis of paused promoters. What is more surprising is that half of the reservoirs also did not produce nascent transcripts within 6Kb of the TSS (ghosts). The distribution of number of DBPs was not

different between poised and ghost promoters. Nor could we find enrichment of specific DBPs that separate these categories. Another possibility is that ghosts are positioned in a three-dimensional space with "DBP" hubs [44,45]. Finally, it could be that the large number of binding events at these promoters causes a 'liquid phase state transition' owing to so many proteins in a confined space.

Our permutation-based approach to determine if a DBP prefers a genomic feature allowed us to extend beyond promoters into the noncoding genome. Specifically, we were interested in determining if certain DBPs were specific to repetitive elements, such as transposons, across the genome. Comparing random permutation versus observed overlaps revealed something somewhat surprising: that repeat classes and families such as 'simple-repeats' and tRNA repeats were strongly enriched for all DBPs tested. In contrast, Line and Satellite repeats were strongly depleted for all DBPs. Thus, some repeat sequences 'repel' DNA binding and some 'recruit' DBPs without discretion.

In some cases, we did observe some interesting biases for DBPs and repeat elements. One example is the human specific repeat family 'SVA' as one of the newest evolving repeats in humans compared to primates. Specifically, three genes had a strong bias of binding SVA elements—all three of which are known transcriptional repressors. Recently studies have identified that primate specific transposons can be co-opted to generate promoters of newly evolving enhancers and even lncRNAs [27,46–48]. Thus, unlike many existing examples of co-option in the case of SVA, it could have selective pressure for binding motifs of the observed repressors and hitherto to unknown repressor motifs–or hitherto unknown promoter regulatory elements.

Collectively, this exercise in data-science, reproducibility and scale in a singular cellular context has been informative to understand relativistic promoter binding events across 161 DBPs. This has led us to understand new features of the coordination of this binding with respect to promoter expression output. Perhaps most importantly, 15 graduate students learned data-sciences and reproducibility measures that not only provide new insight into reservoir promoters but also a logical framework for future objective teaching exercises of genomic data-science.

All markdown files needed to reproduce the results and figures of this manuscript can be found here: https://github.com/boulderrinnlab/CLASS_2020.

## Materials and methods

### ChIP-seq

All ChIP-seq data was downloaded from the encode portal with the selection criterion of 100bp paired end reads, k562 cell line and required replicate. The files meeting this criterion can be found on the ENCODE Portal via this link https://www.encodeproject.org/search/?type=Experiment&status=released&assay_slims=DNA+binding&replicates.library.biosample.donor.organism.scientific_name=Homo+sapiens&biosample_ontology.term_name=K562&assay_title=TF+ChIP-seq&biosample_ontology.classification=cell+line&files.read_length=100&assay_title=Control+ChIP-seq&advancedQuery=date_released:%5b2009-01-01%20TO%202020-01-31%5d.

All corresponding FASTQ files for ChIP and controls were downloaded and used as input into the nextflow pipeline nf-core/chipseq (using version 1.1.0). The resulting broadPeak and BigWig files are available on the UCSC Genome Browser trackHub: http://genome.ucsc.edu/cgi-bin/hgTracks?db=hg38&hubUrl=https://bchm5631sp2020.s3.amazonaws.com/trackhub/hub.txt. Individual broadPeak files were further merged into a "consensus peak" file that merges any peaks that overlap in both data sets. We took the union of these intersections. All

analyses performed on the consensus peaks are outlined in stages of analyses on our github: https://github.com/boulderrinnlab/CLASS_2020. All genomic features were obtained using GENCODE v32 according to the human genome release hg38.

### RNA-seq

Total RNA-seq data from two replicates of K562 total RNA and PolyA+ was obtained from ENCODE: (specifically these file accessions: ENCFF044SJL, ENCFF048ODN, ENCFF381BQZ, ENCFF492ENI, ENCFF625ZBS, ENCFF630HEX, ENCFF671NWM, ENCFF728JKQ). For the gene-level quantification we used Gencode v32 gene annotations and generated read counts from the above BAM files using Rsubread featureCounts (Bioconductor v3.10). For the conservative list of reservoirs, we ran featureCounts using a custom promoter annotation +/-3kb from the Gencode v32 TSS. Here, featureCounts was run with multi-overlap = TRUE to count any read falling withing the promoter region. (https://github.com/boulderrinnlab/CLASS_2020/blob/master/analysis/07_binding_versus_expression/07_binding_vs_expression.Rmd).

### PRO-Seq

Two replicate K562 PRO-seq BAM file data sets were obtained from Arpeggio Biosciences (GEO: GSE166658) and nascent RNA levels were quantified over the +/-3kb promoter windows from the Gencode v32 annotated TSS for lncRNA and protein coding genes. Read counts over these regions were calculated using Bioconductor Rsubread (v3.10) and converted to TPM (https://github.com/boulderrinnlab/CLASS_2020/blob/master/analysis/10_reservoir_nascent_txn/10_reservoir_nascent_txt.Rmd).

### Super enhancer annotations

Annotations of super-enhancers in K562 were obtained from Super-Enhancer DB (SEdb): http://www.licpathway.net/sedb/. The annotations for super-enhancers were in human genome release 19 (hg19) and were lifted over using the UCSC "liftover" tool according to hg19ToHg38.over.chain.gz (https://github.com/boulderrinnlab/CLASS_2020/blob/master/analysis/08_defining_reservoirs/08_defining_reservoirs.Rmd).

### Statistical analyses

For most analyses where ChIP-seq data was compared to genome annotation features from GENCODE v32 we calculated empirical null distributions of overlap using 1,000 random (seeded) permutations followed by fisher-exact test of observed value versus expected distribution. Promoter profile plots were generated by centering windows on the TSS and 3 kilobases up and down stream. We further calculated the 95th percentile of confidence using the variance of average coverage across the promoter window (https://github.com/boulderrinnlab/CLASS_2020/tree/master/analysis/04_promoter_features_profile_plots). We further used quartile analyses to classify several genomic features. The quantile and rationale for each are in the main text and specified R markdown files.

## Supporting information

**S1 Fig.** (A) Distribution of number of consensus peaks observed for each DBP with cutoff at 15th percentile shown as red line. (B) Permutation analysis of DBP significance of overlapping a promoter versus 1,000 random samplings of the same peak profiles for each DBP genome wide. Showing enrichment and depletion status for DBPs (Fisher Exact P < 0.01). (EPS)

**S2 Fig.** UMAP dimensionality reduction based on DBP binding profiles and overlaid with: (A) DNA binding domain annotations. (B) enrichment score on reservoir promoters (C) TF annotation status (D) Median RNA-seq expression level of bound promoters. (E) Examples promoter binding profile. Grey line indicates 95% confidence interval and black line is the mean value. (F) Heatmap of each promoter binding profile for individual DBPs centered at TSS. Red indicates degree of binding. Cluster of binding profiles for each DBP. The four clusters are separated by white space. (G) Enrichment for each DBP at lncRNA and mRNA promoters versus 1,000 random samplings of the same profiles for each DBP across the genome. Blue indicates Z-score of observed versus permuted distribution.
(EPS)

**S3 Fig. DBP binding profile across multiple genomic features.** The fraction of peaks bound is plotted by type of genomic feature (e.g., intergenic space).
(EPS)

**S4 Fig. Heatmaps as in Fig 5 for all SVA elements in the human genome.** (A) Histone modifications (B) DBPs enriched at SVAs. (C) Expression of SVA elements relative to other LTR containing endogenous retroviruses (ERVs).
(EPS)

**S5 Fig.** (A) X-axis, number of DBPs bound per promoter for all promoters. Y-axis is the log10 (TPM) expression of resulting transcript as measured by RNA-seq. (B) Cumulative distribution of binding events on promoters. Red line indicates approximately the 50th percentile of binding events occurring at 7 DBPs bound per promoter. (C) Stacked box plots of lncRNA (red) and mRNA (black) promoters in reservoirs versus non-reservoirs. (D) Stacked box plots of promoter types in reservoir (right) versus non reservoir (left) (E) Bar plot of the 25% of reservoir promoters that have other promoters nearby. True equals a neighboring gene promoter is expressed, False is not expressed. (F) X-axis is Chi-squared test value as log2(observed/ expected), Y-axis is the log10 of Chi-squared P-value. (G) UMAP reduction using only DBP binding to only reservoir promoters.
(EPS)

**S6 Fig. Promoter binding meta-plots for reservoir and non-reservoir promoters.** Similar profile plots as in Fig 3B. Reservoir promoters (red) and non-reservoir (grey) promoter binding profiles are overlayed.
(EPS)

**S7 Fig.** (A) Density plot of nascent expression at reservoirs versus non-reservoirs. (B) Box plot of nascent expression without (left) and with (right) H3K27ac modifications. (C) Same as (B) for H3K27me3. (D-E) Same as (B) for reservoir versus non-reservoir promoters. (F) Boxplot of DBP distribution at ghosts versus non-ghosts.
(EPS)

**S1 Table. Sample information for DNA binding proteins in study.**
(CSV)

**S2 Table. Promoter-level summary of DBP properties examined.** Each observation (row) is a promoter and each column a variable investigated in this study.
(CSV)

**S3 Table. Homer peak annotations of soverlap with genomic features.** This table contains the fraction of peaks (per DBP) overlapping several genomic features.
(CSV)

**S1 File. All scripts used to analyze the data and produce figures.**
(ZIP)

## Acknowledgments

We thank the BioFrontiers IT team for all their support and troubleshooting during the semester. We thank the University of Colorado Boulder Biochemistry department for supporting this course.

## Author Contributions

**Conceptualization:** Michael J. Smallegan, John L. Rinn.

**Data curation:** Michael J. Smallegan, John L. Rinn.

**Formal analysis:** Michael J. Smallegan, Soraya Shehata, Savannah F. Spradlin, Alison Swearingen, Graycen Wheeler, Arpan Das, Giulia Corbet, Benjamin Nebenfuehr, Daniel Ahrens, Devin Tauber, Shelby Lennon, Kevin Choi, Thao Huynh, Tom Wieser, Kristen Schneider, Michael Bradshaw, John L. Rinn.

**Investigation:** Michael J. Smallegan, Joel Basken, Maria Lai, Timothy Read, John L. Rinn.

**Methodology:** Michael J. Smallegan, John L. Rinn.

**Project administration:** Michael J. Smallegan, John L. Rinn.

**Resources:** Graycen Wheeler, Joel Basken, Maria Lai, Timothy Read, Matt Hynes-Grace, Dan Timmons, Jon Demasi, John L. Rinn.

**Software:** Michael J. Smallegan, Soraya Shehata, Savannah F. Spradlin, Alison Swearingen, Arpan Das, Giulia Corbet, Benjamin Nebenfuehr, Daniel Ahrens, Devin Tauber, Shelby Lennon, Kevin Choi, Thao Huynh, Tom Wieser, Kristen Schneider, Michael Bradshaw, John L. Rinn.

**Supervision:** Michael J. Smallegan, John L. Rinn.

**Validation:** Michael J. Smallegan, John L. Rinn.

**Visualization:** Michael J. Smallegan, Soraya Shehata, Savannah F. Spradlin, Alison Swearingen, Graycen Wheeler, Arpan Das, Giulia Corbet, Benjamin Nebenfuehr, Daniel Ahrens, Devin Tauber, Shelby Lennon, Kevin Choi, Thao Huynh, Tom Wieser, Kristen Schneider, Michael Bradshaw, John L. Rinn.

**Writing – original draft:** Michael J. Smallegan, Soraya Shehata, Savannah F. Spradlin, Alison Swearingen, Graycen Wheeler, John L. Rinn.

**Writing – review & editing:** Michael J. Smallegan, John L. Rinn.

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
