## [Decision Letter · Decision Letter 0]

14 Oct 2020

PONE-D-20-22097

Genome-wide binding analysis of 195 DNA Binding Proteins reveals “reservoir” promoters and human specific SVA-repeat family regulation

PLOS ONE

Dear Dr. Smallegan,

Thank you for submitting your manuscript to PLOS ONE. After careful consideration, we feel that it has merit but does not fully meet PLOS ONE’s publication criteria as it currently stands. Therefore, we invite you to submit a revised version of the manuscript that addresses the points raised during the review process.

We look forward to receiving your revised manuscript.

Kind regards,

Roberto Mantovani

Academic Editor

PLOS ONE

Journal Requirements:

'The author(s) received no specific funding for this work.'

We note that one or more of the authors are employed by a commercial company: Arpeggio Biosciences

Reviewers' comments:

Reviewer's Responses to Questions

**Comments to the Author**

1. Is the manuscript technically sound, and do the data support the conclusions?

Reviewer #1: Partly

2. Has the statistical analysis been performed appropriately and rigorously? 

Reviewer #1: Yes

3. Have the authors made all data underlying the findings in their manuscript fully available?

Reviewer #1: Yes

4. Is the manuscript presented in an intelligible fashion and written in standard English?

Reviewer #1: Yes

5. Review Comments to the Author

Reviewer #1: In this paper Shehata et al present a very detailed re-analysis of DNA binding profiles of more than 190 DNAbinding proteins in K562 cell lines, as derived from the ENCODE project. The authors present several confirmatory findings, which corroborate the results of previous studies, but also some original and new interesting observations. The structure of the paper is very rational and the English is good. The manuscript reads very well good. All in all my evaluation of the work by Shetata et al is positive in general, however I some concerns/issues that the authors should address before the publication of their work:

1) Since the authors make a point about the reproducibility of bioinformatics analyses, I believe that apart from sharing the code that was used to execute alla their analyses, they should also make available some intermediate files and results. Including for example bigWig fles with detailed coverage profiles, and/or Peak calling files in bed format. This could be easily done by building a track hub on the UCSC genome browser.

2) Related to 1, since also the ENCODE project has defined a very detailed and reproducible workflow for the analysis of their data (see https://www.encodeproject.org/pipelines/ENCPL138KID/), I believe that the authors could/should present a mode detailed comparison of their results with those obtained/available from ENCODE. What are the main differences between the 2 pipelines? How many peaks are in common? How many are new? Is any systematic/consistent pattern observed

3) While the code as available from their main Github repository is very tidy and well commented, I believe that authors should also write a "proper" material and methods section. For example, by reading the paper it is not exactly clear which assembly of the genome assembly they used in their analysis, and/or also the reference annotation of the genome that was used. More details on the tools and software libraries that were used should also be provided. A scientific paper is usually intended to have a broad scientific audience. The methods should be explained clearly, and be intelligible to everyone, not only to bioinformaticians or people that are proficient in the R programming language.

4) Sometimes authors introduce hard cut-off for the binning of the data. See for example, gene expression data: Low: (0.001,0.137] TPM, how were these cut-off derived? more details should be provided. To be honest I did not understand completely which type of RNA-seq data were used for the estimation of gene expression levels, but also how the data were processed. Were these obtained directly from ENCODE? Please specify

5) The authors should elaborate a little on the concept of "reservoir promoter". This is not explained very clearly in the introduction. And while I can grasp the concept I would like to see a more detailed explanation of possible molecular mechanisms/implications. While I believe that this is a reasonable and fascinating hypothesis, currently it not backed up by experimental evidence. So several sentences in the paper regarding the existence and possible functional roles of reservoir promoters should be toned down in my opinion

6) Do the peaks associated with reservoir promoters have the same features as all the other peaks in the datasets? For example do they have the same width? If you consider the distance distribution from annotated TSSs, does that match? But also do the genes associated with reservoir promoters have "typical" features? In terms of exon size, exon number and/or number of transcripts associated with them. Are these genes supported by different systems for the annotation of the human genome: i.e Gencode and Refseq? If you take large datasets of gene expression across different tissues are these genes expressed? I believe that this type of analyses could help you to understand if some of the patterns that you observe could be associated with biases in reads mapping and/or the annotation of the genome

7) Although maybe not the main point of the paper. To confirm your findings and demonstrate that "reservoir promoters" are real, you should demonstrate that you observe the same patterns (but ideally at also at different loci) in at least another cell line. If additional analyses can not be performed, please tone down and discuss this in more details in the discussion.

8) In general the resolution of the figures is low. I do not know if this is due to some problem/flaws in the conversion of the figures in pdf format in the PLOS-ONE online submission system. But please try to provide figures of higher quality. Some text is hardly intelligible . Maybe you should attach a pre-compiled pdf with all the figures in the revision

9) Having a figure (even a supplementary) and/or a section to show- in general- which type of genomic features and in which proportion are associated with ChIP-seq peaks (gene bodies, promoters, enhancers, intergenic) could be useful. Also for lines 165-180? maybe you could try to see if there are some patterns and if you can classify your DNA binding proteins based on the type of occupancy that they have on the genome

10) Since the majority of reservoir promoters are associated with non coding genes, does the correlation of number of binding events with expression level improve if you exclude these from your analyses?

Anyway, and partially off the records. I teach R and bioinformatics too. I have to say that my class would hardly be capable to to such and impressive amount of work. I am impressed!.

So keep going with the good work. You are almost there

6. PLOS authors have the option to publish the peer review history of their article (what does this mean?). If published, this will include your full peer review and any attached files.

Reviewer #1: No

---

## [Author Response · Author response to Decision Letter 0]

30 Mar 2021

We revised the conflict of interest and financial statement as requested. The response to the reviewer has been uploaded as a part of the manuscript submission with the filename: response_to_reviewer_comments.docx

---

## [Decision Letter · Decision Letter 1]

14 May 2021

Genome-wide binding analysis of 195 DNA Binding Proteins reveals “reservoir” promoters and human specific SVA-repeat family regulation

PONE-D-20-22097R1

Dear Dr. Smallegan,

We’re pleased to inform you that your manuscript has been judged scientifically suitable for publication and will be formally accepted for publication once it meets all outstanding technical requirements.

Kind regards,

Roberto Mantovani

Academic Editor

PLOS ONE

Additional Editor Comments (optional):

Reviewers' comments:

Reviewer's Responses to Questions

**Comments to the Author**

1. If the authors have adequately addressed your comments raised in a previous round of review and you feel that this manuscript is now acceptable for publication, you may indicate that here to bypass the “Comments to the Author” section, enter your conflict of interest statement in the “Confidential to Editor” section, and submit your "Accept" recommendation.

Reviewer #1: All comments have been addressed

2. Is the manuscript technically sound, and do the data support the conclusions?

Reviewer #1: Yes

3. Has the statistical analysis been performed appropriately and rigorously? 

Reviewer #1: Yes

4. Have the authors made all data underlying the findings in their manuscript fully available?

Reviewer #1: Yes

5. Is the manuscript presented in an intelligible fashion and written in standard English?

Reviewer #1: Yes

6. Review Comments to the Author

Reviewer #1: The authors have addressed all the main points of criticism raised by this reviewer in a consistent and thorough manner. In my opinion this is a very interesting paper. And it should be accepted

7. PLOS authors have the option to publish the peer review history of their article (what does this mean?). If published, this will include your full peer review and any attached files.

Reviewer #1: No

---

## [Editor Report · Acceptance letter]

16 Jun 2021

PONE-D-20-22097R1 

Genome-wide binding analysis of 195 DNA Binding Proteins reveals “reservoir” promoters and human specific SVA-repeat family regulation 

Dear Dr. Smallegan:

I'm pleased to inform you that your manuscript has been deemed suitable for publication in PLOS ONE. Congratulations! Your manuscript is now with our production department. 

Kind regards, 

on behalf of

Prof. Roberto Mantovani 

Academic Editor

PLOS ONE